# Land Use and Land Cover Classification in the Northern Region of Mozambique Based on Landsat Time Series and Machine Learning

Lucrêncio Silvestre Macarringue [1,2,*], Édson Luis Bolfe [1,3], Soltan Galano Duverger [4], Edson Eyji Sano [5], Marcellus Marques Caldas [6], Marcos César Ferreira [1], Jurandir Zullo Junior [7] and Lindon Fonseca Matias [1]

1   Institute of Geosciences, State University of Campinas (UNICAMP), Campinas 13083-855, Brazil; ebolfe@unicamp.br or edson.bolfe@embrapa.br (É.L.B.); macferre@unicamp.br (M.C.F.); lindon@unicamp.br (L.F.M.)
2   Department of Research, Instituto Politécnico de Ciências da Terra e Ambiente, Matola P.O. Box 58, Mozambique
3   Brazilian Agricultural Research Corporation (Embrapa Agricultura Digital), Campinas 13084-886, Brazil
4   Programa de Pós-Graduação em Difusão do Conhecimento (PPGDC), Universidade Federal da Bahia (UFBA), Salvador 40110-909, Brazil; solkan1201@gmail.com
5   Brazilian Agricultural Research Corporation (Embrapa Cerrados), Planaltina 73301-970, Brazil; edson.sano@embrapa.br
6   Department of Geography and Geospatial Sciences, Kansas State University, Manhattan, KS 66503, USA; caldasma@ksu.edu
7   Center for Meteorological and Climatic Research in Agriculture (CEPAGRI), University of Campinas (UNICAMP), Campinas 13083-889, Brazil; jurandir@cpa.unicamp.br
*   Correspondence: lucrencio.macarringue@gmail.com

**Abstract:** Accurate land use and land cover (LULC) mapping is essential for scientific and decision-making purposes. The objective of this paper was to map LULC classes in the northern region of Mozambique between 2011 and 2020 based on Landsat time series processed by the Random Forest classifier in the Google Earth Engine platform. The feature selection method was used to reduce redundant data. The final maps comprised five LULC classes (non-vegetated areas, built-up areas, croplands, open evergreen and deciduous forests, and dense vegetation) with an overall accuracy ranging from 80.5% to 88.7%. LULC change detection between 2011 and 2020 revealed that non-vegetated areas had increased by 0.7%, built-up by 2.0%, and dense vegetation by 1.3%. On the other hand, open evergreen and deciduous forests had decreased by 4.1% and croplands by 0.01%. The approach used in this paper improves the current systematic mapping approach in Mozambique by minimizing the methodological gaps and reducing the temporal amplitude, thus supporting regional territorial development policies.

**Keywords:** Google Earth Engine; deforestation; feature selection; miombo; random forest

## 1. Introduction

Approximately 70% of Mozambique, located in southeastern Africa and occupying an area of about 800,000 km$^2$, is covered by forestlands and woodlands. According to Chamberlin et al. [1], Mozambique has about 214,000 km$^2$ of non-forested lands potentially suitable for croplands. Miombo, the main forest type in the country, occupies most of the northern and central regions, accounting for approximately two-thirds of the total area of 800,000 km$^2$, ranging from the Rovuma River in the extreme north to the Limpopo River in the south [2–4]. Miombo is more common in the northern region, covering vast areas of Niassa, Nampula, and Cabo Delgado provinces. It is also the predominant biome in neighboring countries of Tanzania, the Democratic Republic of Congo, Angola, Zambia, Malawi, and Zimbabwe [5,6]. This biome is the source of subsistence for the vast majority of rural populations (~66%) [7], providing charcoal, medicinal plants, and areas for religious cults, among other services.

Systematic land use and land cover (LULC) surveys are crucial for ensuring sustainable use of natural resources, given the increasing degradation that occurs in the region, combined with illegal selective logging due to inefficient environmental law enforcement and frequent uncontrolled fires and land clearings by small family farms (*machambas*) [8]. Mozambique has faced serious challenges in performing this mapping at the regional level and, consequently, in managing land resources effectively. Studies on LULC changes in Mozambique are incipient. Previous reports have addressed land-cover changes in the Limpopo River Basin [9], a woodland-based ecosystem service in Mabalane [10], assessment of LULC changes, biodiversity and land management in Quirimbas National Park [11], and mapping smallholder and large-scale cropland dynamics in the emerging frontier of Mozambique [12], among others issues. These studies did not produce large-scale reference data for spatial and temporal analysis of vegetation types of the country. In other words, studies using large-scale mapping methodology to produce national LULC data are limited in Mozambique. In the four decades of independent Mozambique, only four LULC surveys have been conducted—in 1980, 1994, 2007, and 2018. In addition to the long time intervals between surveys, different methodologies were used in each because of institutional and political issues, such as lack of qualified personnel, limited budgets, interference in the administration of government institutions, and the civil war.

Mozambique's highly complex landscapes make mapping difficult, mostly because of their fragmentation driven by deforestation and forest degradation that is linked with the displacement of agriculture and expansion of residential areas. These factors act in multiple and complex ways [13] and threaten the ecological stability of the ecosystems. One of the issues that contribute to this scenario is the characteristics of rural Mozambique´s landscapes, which host approximately 67% of the national population in scattered settlements. Another factor is the small-scale family farming, which is the main occupation of most of the population, especially in the northern region. The typical size of these farms lands is around 2 ha [14]. As a rainfed and polyculture system, family farming ultimately generates high LULC dynamics, which is included as one of the main causes of forest degradation [13], making its detection by sensors operating with moderate spatial resolution difficult.

The development of different classification algorithms and the overall policy of free distribution of moderate- to coarse-spatial-resolution remote-sensing products has opened up new possibilities for producing LULC maps [15]. This is the case, for example, in the development of neural-network-based machine learning (ML) algorithms in cloud-computing environments, such as the Google Earth Engine™ (GEE) platform. GEE and other similar platforms (e.g., Amazon Web Services™) allow large volumes of data processing on a planetary scale and at high spatial resolution [16–19], overcoming the limitations of the traditional approach in terms of computational speed and costs.

Wang et al. [20] stated that time-series data processed in the GEE platform allows for the development of an effective sample migration approach, that is, the production of long-term and large-scale LULC maps through extracting unchanged samples from the time series by temporal analysis. Efforts have been made to standardize LULC classification processes [21], but each country has its own specificities of land occupation, requiring national-based procedures. In Mozambique, there are few regional-scale LULC databases that can be compared with each other. Based on time series, multi-resolution fusion, and multi-source data from Landsat series, we hypothesized that LULC mapping across regional or national scales can be completed quickly and accurately.

Taking advantage of the cloud-computing capabilities of GEE, and the data dimensionality reduction techniques, this study aimed to map LULC classes of the northern region of Mozambique between 2011 and 2020 based on Landsat time series processed by an ML classification approach.

## 2. Materials and Methods

### 2.1. Study Area

The northern region of Mozambique encompasses the Niassa, Cabo Delgado, and Nampula provinces, totaling 56 districts: 16 in Niassa, 17 in Cabo Delgado, and 23 in Nampula. This region covers an area of 293,287 km², accounts for ~37% of the national territory, and is bordered by the Republic of Tanzania (north), Zambezia province (south), the Indian Ocean (east), and the Republic of Malawi (west) (Figure 1). It has a population of approximately 10 million inhabitants, about one-third of the country's total population [7].

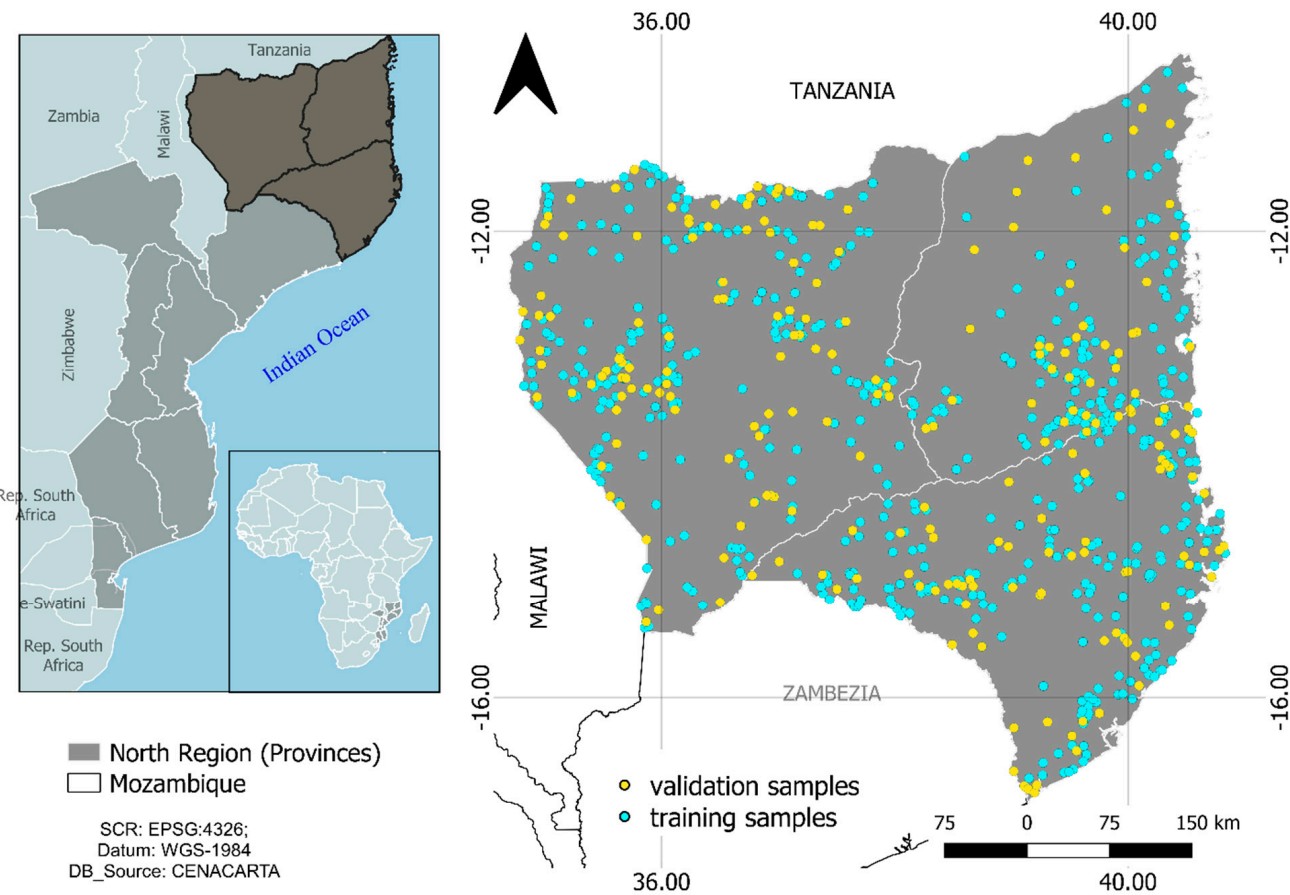

**Figure 1.** Location of the study area in the Mozambique, showing training samples (cyan) and validation samples (yellow) collected in 2020. Source: database from the National Cartography and Remote Sensing Centre (CENACARTA).

A humid tropical climate prevails in the northern region of Mozambique, with a rainy and hot season from October to April, and a cool and dry season from May to September [22–24]. The annual average temperature ranges from 25 °C to 26 °C in the coastal areas of the provinces of Nampula and Cabo Delgado and tends to decrease inland, reaching 22 °C in Lichinga (Niassa). The terrain is gently undulating with flat sections, where sparse, residual mountainous elevations are found. The dominant soil types are Eutric Cambisols, Ferric Lixisols, and dune sediments in the coastal plain and Red Haplic Acrisols and Rhodic Ferralsols, in addition to the Calcic Vertisols further inland [23].

### 2.2. Data Acquisition

LULC was mapped from the top-of-the-atmosphere (TOA) reflectance data retrieved from the time series of the Landsat 7, Enhanced Thematic Mapper Plus (ETM+) (LANDSAT/LE07/C01/T1_RT_TOA) and Landsat 8, and Operational Land Imager (OLI) (LANDSAT/LC08/C01/T1_RT_TOA) images from 2011 to 2020, available in the GEE platform.

This period was defined based on the high levels of deforestation alerts that were broadcast by press, civil society, and the scientific community.

The study area was covered by 24 Landsat scenes with a repeat pass of 16 days (paths from 164 to 168; rows from 67 to 72). We selected a set of 528 TOA reflectance Landsat images without cloud coverage and with good radiometric quality, which were used as a baseline for the classification scheme. The band designations, descriptions, and resolutions are summarized in Table 1.

**Table 1.** Characteristics of Landsat 7 Enhanced Thematic Mapper Plus (ETM+) and Landsat 8 Operational Land Imager (OLI) optical images used in this study. NIR = near infrared; SWIR = shortwave infrared; PAN = panchromatic.

| Resolution | Landsat 7 ETM+ | | | | Landsat 8 OLI | | |
|---|---|---|---|---|---|---|---|
| | Band | Spectra | Wavelength (μm) | | Band | Spectra | Wavelength (μm) |
| Spectral resolution | 1 | Blue | 0.45−0.52 | Spectral resolution | 2 | Blue | 0.45−0.51 |
| | 2 | Green | 0.52−0.60 | | 3 | Green | 0.53−0.59 |
| | 3 | Red | 0.63−0.69 | | 4 | Red | 0.64−0.67 |
| | 4 | NIR | 0.77−0.90 | | 5 | NIR | 0.85−0.88 |
| | 5 | SWIR 1 | 1.55−1.75 | | 6 | SWIR 1 | 1.57−1.65 |
| | 7 | SWIR 2 | 2.09−2.35 | | 7 | SWIR 2 | 2.11−2.29 |
| | 8 | PAN | 0.52−0.90 | | 8 | PAN | 0.50−0.68 |
| Temporal resolution | 16 days | | | | 16 days | | |
| Radiometric resolution | 8 bits | | | | 12 bits (scaled to 16 bits) | | |
| Spatial resolution | 30 m | | | | 30 m | | |

*2.3. Methods*

The time series of Landsat scenes (ETM+ for the years 2011 and 2012 and OLI for the remaining years) were processed in the GEE platform, which provides cloud-computing capability and global catalog services of various sensors such as Landsat, Sentinel-1, and Sentinel-2 satellites, and various global land-cover and climate datasets, among others [25–27]. The time series were filtered to select images with cloud coverage below 70% and stored in the GEE platform. The filtered collection was reduced to a single median image per year to minimize the impact of clouds, cloud shadows, and noise.

We performed a band fusion of panchromatic and multispectral bands using the Gram–Schmidt algorithm [28]. The imagery was then subset to the area of interest. A set of 22 spectral indices, including the three tasseled-cap-transformation bands, was generated to be included in the classification process. We also performed a feature-selection technique based on the random forest (RF) approach to avoid data redundancy.

A post-classification statistical analysis is presented using the concepts presented in the literature (Kappa, overall accuracy, F1-score, and user and producers accuracy) [29], Phyton programming language, and semi-automatic classification plug-in (SCP) in the QGIS image-processing software. Figure 2 summarizes the methodological steps of image processing used in this study.

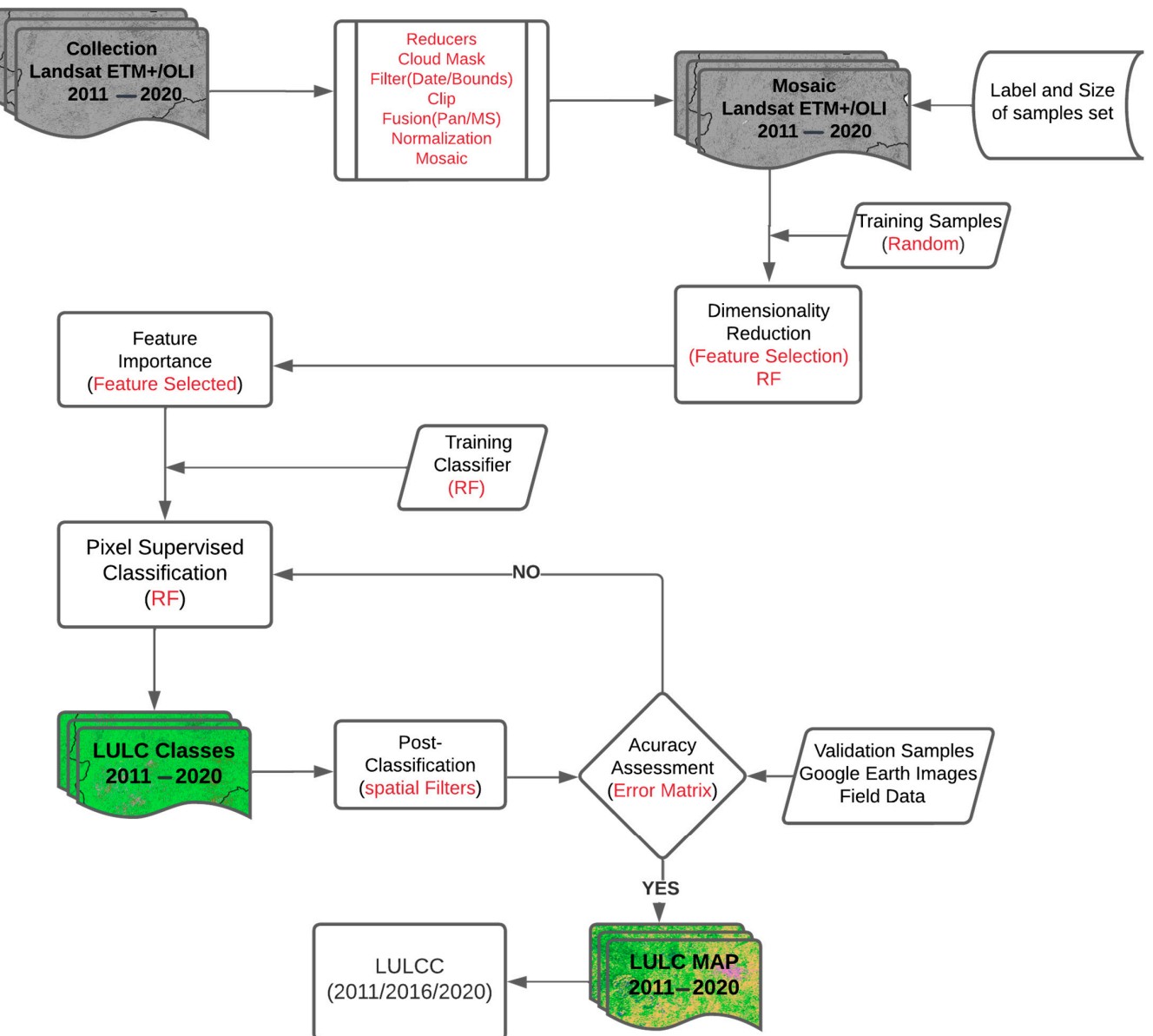

**Figure 2.** Flowchart of the methodological steps of image processing used in this study. LULC = land use and land cover; RF = random forest.

### 2.3.1. LULC Classes and Samples

LULC mapping based on moderate resolution satellite images in Mozambique is not an easy task because of the highly fragmented landscape [30,31]. For example, in the last national forest inventory conducted between 2015 and 2017, MITADER [3] highlighted difficulties in inventorying coastal forests and mangroves, differentiating Miombo woodlands from semi-deciduous forests, and distinguishing agricultural areas from grass/shrub covers as well as residential areas from non-vegetated areas. We chose a polygon as a sample unit applying a dichotomous key with binary choice (yes or no) for class label [29].

In this paper, five LULC classes were defined (Table 2): dense vegetation, open evergreen and deciduous forests, non-vegetated areas, croplands, and built-up areas. Most agricultural areas in Mozambique span 2 ha on average, accounting for approximately 95% of the total cultivated area across the country. About 34% of them are found in the northern part of the country [14] and agriculture is mostly based on rainfed farming. Urban areas are poorly structured, with few vertical buildings (up to four floors, in general). Most urban

areas present unpaved roads and scattered horizontal buildings aligned with the main paved roads [31].

**Table 2.** Description of the land-use and land-cover classes considered in this research and the number of samples collected for classification.

| Number of Samples (Points) * | Sample Separability | Spectral Class | Field Photo |
|---|---|---|---|
| 129,240 |  | Dense Miombo forests, mangroves, gallery forests, and planted forests. Minimum area of 1 ha, with native and exotic semi-perennial trees with at least 3 m height and canopy cover $\geq$ 30% (mangroves). Transition lands between marine and terrestrial environments. Altitude varying from sea level to 1000 m. |  |
| 201,021 |  | Open evergreen and deciduous forests. Including grasslands (native pastures and grasses, weeds in flooded areas). Minimum area of 1 ha, temporarily or permanently flooded. |  |
| 61,859 |  | Non-vegetated areas (dunes, beaches, and fallow lands). |  |
| 11,477 |  | Croplands. Average size of 2 ha, family farming, and subsistence/irrigation practices. Include both temporary and permanent crops. |  |
| 34,711 |  | Built-up areas (cities, towns, and villages). Low, medium, or high population density, poorly structured. Mostly located along the main access roads. |  |

Source: https://mozrealblog.wordpress.com/2017/11/07/vereadores-e-chefes-locais-discutem-lugares-na-cidade-de-nampula/. Accessed on 30 January 2022. * Samples relate to the year 2020.

The quality and quantity of training samples are among the most important factors affecting the accuracy of classification results [29,32–34]. Different sampling techniques

(for example, random, or systematic) and criteria (for example, field survey, crowdsourcing, or visual inspection) of training and validation data-collection are presented in the literature [29,32,35].

In this paper, field surveys and visual interpretation of high-spatial-resolution images available in the Google Earth Pro and GEE were used as sampling methods to help identify both training and validation samples for the year 2020. The samples used in the year 2020, acquired through field surveys and high-resolution satellite images from Google Earth, were used as the basis for identifying the labels of the samples from other years, verifying whether there was a change in the label or not (sample migration approach). Thus, the samples for each year were obtained with few differences in size. In the field, the samples were collected using a Garmin Global Positioning System (GPS) receiver, with an accuracy of ~4 m.

The field survey, conducted in April 2020, allowed us to define the mapping classes, collect samples for training and validation, and identify the distribution patterns and characteristics of landscape elements. A total of 438,308 sampling points were collected randomly for the year 2020 of which 70% (306,816 points) were used as the training set and 30% (131,492 points) as the validation set (Figure 1). The number of sampling points for each LULC class (Table 2) was defined considering the distribution pattern of the selected classes and the non-correlation criterion, as recommended by Congalton and Green [29] and Lei et al. [35].

### 2.3.2. Feature Selection (FS) Method

In this paper, we used the RF algorithm for both FS and for the classification. RF has been widely used for similar studies [18,33,36,37]. This algorithm solves regression and classification problems using ensemble learning. It builds multiple decision trees and merges them together to obtain a more accurate and stable prediction [38]. These procedures produce higher accuracy when analyzing complex data on large areas [39] because they are independent of parametric statistical assumptions and, therefore, are better suited for analyzing multimodal, noisy, or missing data and combinations of categorical and continuous ancillary data [40].

For FS, we also used the RF-based approach, an embedded method that provides a variable-importance criterion for each feature and calculates the average decrease of classification accuracies based on a set of non-tuned decision trees. Each tree was increased to its maximum size using a different bootstrap sample of two-thirds of the original training data, with the remaining third retained as "out-of-bag" (OOB) samples used as an internal error estimate of the overall classification accuracy, considering a random number of variables in each split [41–43].

The two user-defined hyperparameters for random-forest models are the number of trees to be generated by the model (ntrees) and the number of randomly selected variables to be used in each tree (mtry). Various values were tested and the accuracy of the final classification result measured for the sample set. This test was also carried out with the help of Python on the colab platform and on the set of samples mentioned above.

To analyze the impact of FS on classification accuracy, the number of variables was fixed at 28 and then reduced systematically to the 10 most important variables. The relative importance of each variable was based on the training samples to generate a random-forest model, with different trees sizes and different variables per split. We relied on the Scikit-Learn module [44], which determined the variable importance by analyzing how much the nodes that use each variable reduced the impurity across all trees on average by weighting the number of training samples reaching each node [45]. The importance of a feature was computed as the normalized total reduction of the criterion brought by that feature, also known as the Gini importance or decrease in Gini impurity (DGI) [45].

Running RF requires only two parameters, the number of trees to be formed (ntree), and the number of variables that are randomly selected to determine the splits at each node of individual trees (mtry) [46]. The first comprised testing the model with a different

number of trees, and the other testing with different numbers of variables and different sizes of classes (5 and 6). The number of ntrees was searched between 35 and 500 using a random interval, whereas the optimal mtry was fixed at 25. Twenty-eight variables were used in the model, including 6 spectral bands and 22 spectral indices (Table 3), calculated based on aggregated data and the median-reducer method, which leads to better results than those produced by original time series [47].

**Table 3.** List of spectral indices used in the image classification.

| Name | Acronym | Formula | Citation |
|---|---|---|---|
| Automated water extraction index | AWEI | $4 \times (Green - SWIR2) - (0.25 \times NIR + 2.75 \times SWIR1)$ | [48] |
| Canopy chlorophyll content index | CCCI | $\dfrac{\left[\frac{NIR - Red_{EDGE}}{NIR + Red_{EDGE}}\right]}{\left[\left(\frac{NIR - Red}{NIR + Red}\right)\right]}$ | [49] |
| Difference vegetation index | DVI | $NIR - Red$ | [50] |
| Enhanced vegetation index | EVI | $2.5 \times \dfrac{NIR - Red}{NIR + 6 \times Red - 7.5 \times Blue + 1}$ | [51] |
| Global environmental monitoring index | GEMI | $(n \times (1 - 0.25 \times n) - [(Red - 0.125)/(1 - Red)];$ $n = \left[\left(2 \times \left(NIR^2 - Red^2\right)\right) + 1.5 \times NIR + 0.5 \times Red\right] / (NIR + Red + 0.5)$ | [52] |
| Global vegetation moisture index | GVMI | $\dfrac{[(NIR + 0.1) - (SWIR1 + 0.02)]}{[(NIR + 0.1) + (SWIR1 + 0.02)]}$ | [53] |
| Indicative index of water bodies | IIA | $\dfrac{Green - 4 \times NIR}{Green + 4 \times NIR}$ | [54] |
| Isoil | - | $\dfrac{NIR - Green}{NIR + Green}$ | - |
| Leaf chlorophyll index | LCI | $\dfrac{NIR - Red}{NIR + Red}$ | [55] |
| Land surface water index | LSWI | $\dfrac{NIR - SWIR1}{NIR + SWIR1}$ | [56] |
| Modified normalized difference water index | mNDWI | $\dfrac{Green - SWIR1}{Green + SWIR1}$ | [57] |
| Moisture stress index | MSI | $\left(\dfrac{SWIR1}{NIR}\right)$ | [58] |
| Normalized difference vegetation index | NDVI | $(NIR - Red)/(NIR + Red)$ | [59] |
| Normalized difference water index | NDWI | $\dfrac{Green - NIR}{Green + NIR}$ | [60] |
| Renormalized difference vegetation index | RDVI | $NIR - \dfrac{Red}{\sqrt{NIR + Red}}$ | [61] |
| Soil-adjusted vegetation index | SAVI | $\dfrac{NIR - Red}{NIR + Red + L} \times (1 + L); L = 0.5$ | [62] |
| Modified soil-adjusted vegetation index | mSAVI | $\dfrac{\left[(2 \times NIR + 1) - \sqrt{(2 \times NIR + 1)^2 - 8 \times (NIR - Red)}\right]}{2}$ | [63] |
| Green soil-adjusted vegetation index | GSAVI | $[(NIR - Green)]/[(NIR + Green + L)] \times (1 + L); L = 0.5$ | [64] |
| Optimized soil-adjusted vegetation index | OSAVI | $(NIR - Red)/(0.16 + NIR + Red)$ | [65] |
| Tasselled cap—vegetation | GVI | $0.1509 \times Blue \quad +0.1973 \times Green + 0.3279 \times Red$ $+0.3406 \times NIR + 0.7112 \times SWIR1$ $+0.4572 \times SWIR2$ | [66,67] |
| Tasselled cap—wetness | WET | $0.1511 \times Blue \quad +0.1973 \times Green + 0.3283 \times Red$ $+0.3407 \times NIR + (-0.7117) \times SWIR1$ $+(-0.4559) \times SWIR2$ | [66,67] |
| Tasselled cap—brightness | SBI | $0.3037 \times Blue \quad +0.2793 \times Green + 0.4743 \times Red$ $+0.5585 \times NIR + 0.5082 \times SWIR1$ $+0.1863 \times SWIR2$ | [66,67] |

The iteration with the best results for the model was used for classification based on the measure of mean decreased accuracy [68]. Confusion matrices were calculated to help select the optimal model in the process of selecting variables using test samples.

2.3.3. LULC Classification and Accuracy Assessment

Although RF can handle a large number of variables and a relatively low number of observations, reducing the number of variables may be advantageous for reducing computation time and/or for determining which variables are considered irrelevant in predicting the characteristic of interest [42]. Compared with other classifiers, RF offers several advantages such as: efficient implementation; higher classification accuracy compared with traditional techniques; a computationally cost-effective technique; random selection of the best split nodes among the subset of predictions instead of splitting it among all variables; and measurement of each variable's contribution to the classification output, which is critical in assessing the value of each variable. Besides, it belongs to the collection of tree-structured classification methods [36,38,69].

In this study, a pixel-based supervised RF algorithm was used for classification. LULC classification was performed in the GEE platform using an RF with 100 trees and 25 nodes, based on the best result of the model. Finally, a 9 × 9 post-classification majority filter was applied to reduce the salt-and-pepper effects on the output classification map. This procedure was carried out utilizing the semi-automatic classification plugin (SCP) available in QGIS software.

Accuracy was assessed based on overall accuracy, which expresses how close the classification is to the reference data [43,47], and user accuracy (UA), producer accuracy (PA), and Kappa index. Another accuracy measure used in this paper was the F1-score (Equation (1)), which combines UA and PA into a single measure [70].

$$F1\text{-}score = \frac{2 \times PA_i \times UA_i}{PA_i + UA_i} \tag{1}$$

where F1-score is the harmonic mean of the two accuracies and PA and UA are the producer accuracy and user accuracy, respectively, for each class *i*.

## 3. Results

### 3.1. Feature Selection and Importance of the Variables

Figure 3 shows the accuracies obtained by this study in terms of tree size and number of variables. This step was performed using training data. Using an RF to select variables not only generates input spectral data for LULC classification and for assessing how this procedure affects classification accuracy, but also considerably decreases the number of variables used by the classifier while simultaneously generating classification accuracies. The final analysis was carried out for the years 2011, 2014, 2016, 2018, and 2020 based on tree sizes (Figure 3A) and number of variables (Figure 3B) and Figure 3C shows the performance of the model in the time series.

Despite significant differences in accuracy over the years, the variation in accuracy was much lower when alternating the number of trees, showing stability after 100 trees. When the number of variables was alternated, the scenario was similar to the previous finding in terms of accuracy over the years. The model became stable after 15 variables. Therefore, the optimal number of trees should be equal to or higher than 100 and with at least 15 variables. Six variables were common to all years (NDVI, LSWI, MSI, NDWI, GVMI, and DVI) and their scores displayed a decreasing trend.

The number of variables for classification was set to 10 because, in addition to improving the classification performance, there was no statistically significant difference in the classification between this number and the one suggested by the model. Therefore, the classification was performed with 100 trees, 25 nodes, and 10 variables. The best accuracy of the model was observed in 2020 (~0.96%) and the worst in 2013 (~0.90%). These accuracies were considered satisfactory to run the classification.

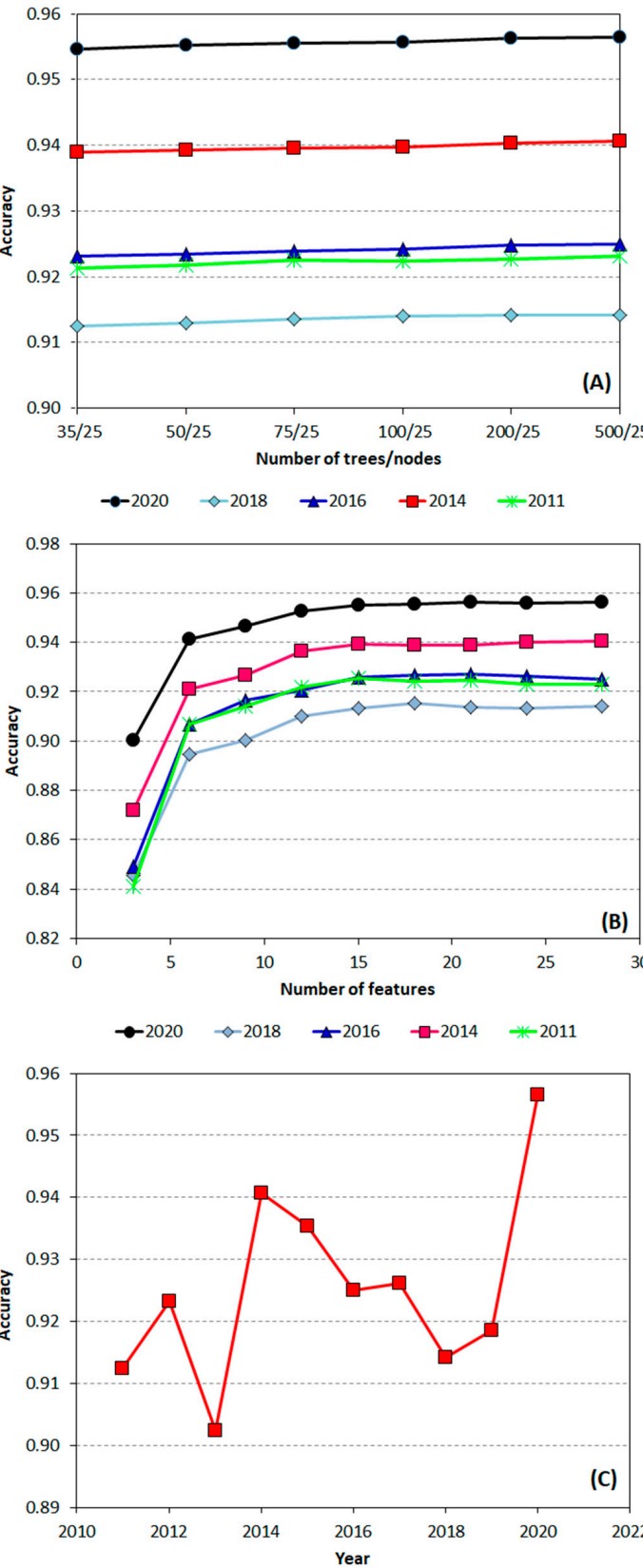

**Figure 3.** Accuracies based on size of trees (**A**), number of variables (**B**), and observed accuracy (**C**) that best explained the model in each year.

The tests performed with different sample sizes revealed that the combination of five classes (dense vegetation, open evergreen and deciduous forest, non-vegetated areas, croplands, and built-up areas) provided the best model-calibration results. The differences were analyzed based on confusion matrices. Although the procedure was repeated for all years, only the results of 2016 and 2019 are shown here for illustration purposes (Table 4).

**Table 4.** Confusion matrices calculated from the training samples in three scenarios using the model. Scenario A = five classes, without grasslands (GL); scenario B = five classes, without croplands (CL); and scenario C = all six classes.

| 2016 (A) (Accuracy = 0.925) | | | | | | 2020 (A) (Accuracy = 0.956) | | | | | |
|---|---|---|---|---|---|---|---|---|---|---|---|
| NVA | 0.93 | 0.04 | 0 | 0.04 | 0 | NVA | 0.95 | 0.03 | 0 | 0.02 | 0 |
| BA | 0.09 | 0.87 | 0 | 0.04 | 0 | BA | 0.07 | 0.9 | 0 | 0.03 | 0 |
| CL | 0 | 0 | 0.62 | 0.08 | 0.3 | CL | 0 | 0 | 0.63 | 0.09 | 0.28 |
| OEDF | 0.01 | 0 | 0 | 0.95 | 0.04 | OEDF | 0 | 0 | 0 | 0.97 | 0.02 |
| DV | 0 | 0 | 0.01 | 0.06 | 0.93 | DV | 0 | 0 | 0.01 | 0.04 | 0.95 |
| | NVA | BA | CL | OEDF | DV | | NVA | BA | CL | OEDF | DV |

| 2016 (B) (Accuracy = 0.815) | | | | | | 2020 (B) (Accuracy = 0.810) | | | | | |
|---|---|---|---|---|---|---|---|---|---|---|---|
| NVA | 0.94 | 0.03 | 0.01 | 0 | 0.02 | NVA | 0.94 | 0.03 | 0.01 | 0 | 0.02 |
| BA | 0.07 | 0.85 | 0.02 | 0 | 0.05 | BA | 0.07 | 0.8 | 0.02 | 0 | 0.05 |
| OEDF | 0 | 0 | 0.69 | 0.06 | 0.25 | OEDF | 0 | 0 | 0.69 | 0.06 | 0.25 |
| DV | 0 | 0 | 0.08 | 0.92 | 0.01 | DV | 0 | 0 | 0.08 | 0.93 | 0.01 |
| GL | 0 | 0 | 0.19 | 0.01 | 0.8 | GL | 0 | 0 | 0.17 | 0.01 | 0.95 |
| | NVA | BA | OEDF | DV | GL | | NVA | BA | OEDF | DV | GL |

| 2016 (C) (Accuracy = 0.807) | | | | | | | 2020 (C) (Accuracy = 0.804) | | | | | | |
|---|---|---|---|---|---|---|---|---|---|---|---|---|---|
| NVA | 0.86 | 0.03 | 0 | 0.01 | 0 | 0.1 | NVA | 0.94 | 0.03 | 0 | 0.01 | 0 | 0.02 |
| BA | 0.06 | 0.83 | 0 | 0.01 | 0 | 0.1 | BA | 0.07 | 0.86 | 0 | 0.02 | 0 | 0.06 |
| CL | 0 | 0 | 0.62 | 0.07 | 0.03 | 0 | CL | 0 | 0 | 0.59 | 0.18 | 0.19 | 0.04 |
| OEDF | 0 | 0 | 0 | 0.79 | 0.04 | 0.16 | OEDF | 0 | 0 | 0 | 0.69 | 0.06 | 0.25 |
| DV | 0 | 0 | 0.01 | 0.06 | 0.93 | 0 | DV | 0 | 0 | 0.01 | 0.08 | 0.91 | 0.01 |
| GL | 0.02 | 0.01 | 0 | 0.27 | 0.01 | 0.7 | GL | 0 | 0 | 0 | 0.18 | 0.01 | 0.8 |
| | NVA | BA | CL | OEDF | DV | GL | | NVA | BA | CL | OEDF | DV | GL |

NVA = non-vegetated areas; BA = built-up areas; CL = croplands; OEDF = open evergreen and deciduous forest; and DV = dense vegetation.

In this figure, we used three scenarios to analyze the effect of different sizes of LULC class due to the impact of this effect on the classification process. Scenario A showed the best performance in the model, reaching an accuracy of approximately 0.92% even though croplands was confused with both open evergreen and deciduous forest and dense vegetation. In scenario B, grasslands was confused with OEDF and presented significant omission and commission errors, affecting its accuracy. In scenario C, croplands and two native vegetation classes were again confused, leading to low accuracy. Based on these scenarios, we chose to work with scenario A which, in addition to providing better performance of the model, presented a balance between LULC classes. The variables were selected and categorized in order of importance (Figure 4).

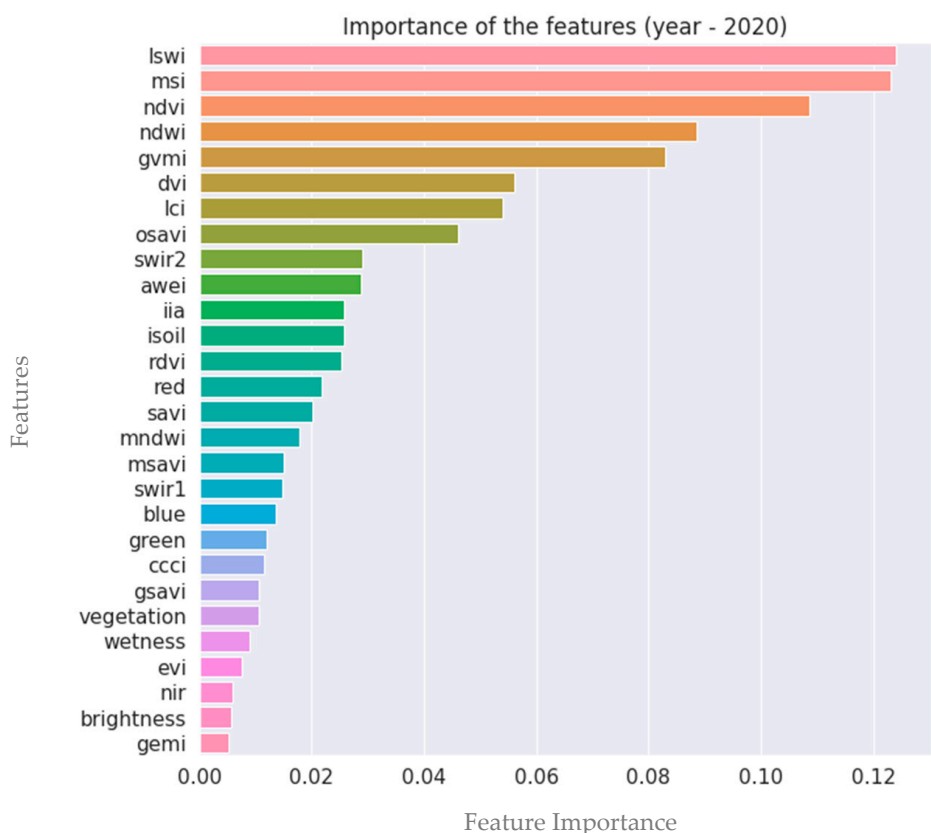

**Figure 4.** Importance of the variables of the year 2020 based on mtree = 500 and mtry = 25 for illustration purposes.

The number of variables considered relevant and suitable remained reasonably consistent after several tests. The 10 most important variables were selected for each year, but only one year is shown for illustration purposes. In these 10 years, the following common variables were selected: NDVI, LSWI, MSI, NDWI, GVMI, and DVI, which were ranked differently in the order of importance. These spectral indices, particularly the NDVI and NDWI, are consistently identified in several LULC mapping initiatives [40,47,71]. Although water bodies were not mapped in this study, the NDWI was shown to be relevant, meaning that it has potential to be considered in future mappings of this region [72,73].

### 3.2. Separability of the Classes

Figure 5 shows the scatterplots between variables and LULC classes used in this research, highlighting, in the diagonal, all classes having intersections, resulting in classification confusion. In general, the discrimination was best between non-vegetated areas and built-up areas and worst between croplands and dense vegetation.

### 3.3. Accuracy of the Maps

The accuracy analysis of the RF classification was developed on training data for the following years: 2011, 2014, 2016, 2018, and 2020. These years were chosen because they illustrated better the dynamics of the landscape for the elements considered. The overall accuracy ranged from 80.5% to 88.7% while the Kappa indices ranged from 0.65 to 0.80 (Table 5). Low PAs were observed for built-up areas and croplands, as shown by the F1-score values. Other studies [19,74,75] also obtained global accuracies close to these values.

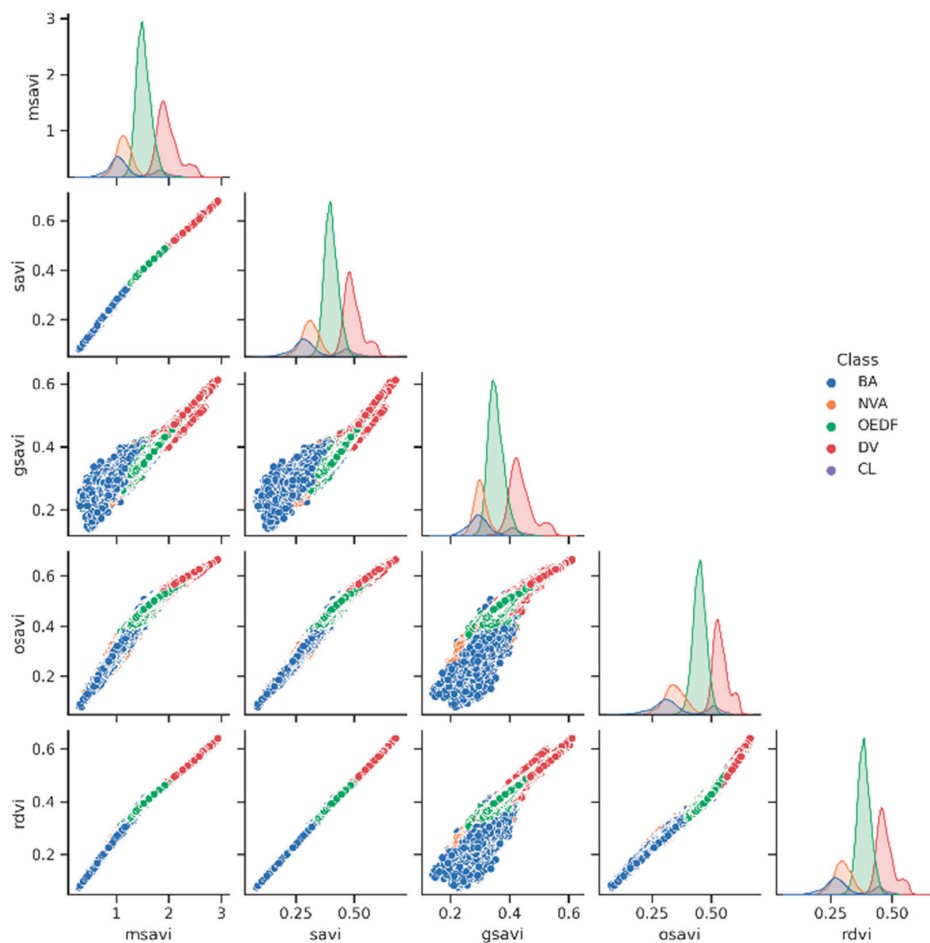

**Figure 5.** Scatterplots showing the relationship between the variables and the spatial separability of the classes mapped for 2020. BA = built-up areas; NVA = non-vegetated areas; OEDF = open evergreen and deciduous forest; DV = dense vegetation; and CL = croplands.

**Table 5.** Accuracies of the random forest classification results in the selected study area.

| Metric/Class | NVA | BA | CL | OEDF | DV | Kappa | OA |
|---|---|---|---|---|---|---|---|
| | | | 2011 | | | | |
| UA | 80.0 | 82.0 | 50.0 | 92.0 | 75.0 | | |
| PA | 92.0 | 47.0 | 43.0 | 87.0 | 85.0 | 0.74 | 84.69 |
| F1-score | 85.0 | 59.0 | 46.0 | 89.0 | 80.0 | | |
| | | | 2014 | | | | |
| UA | 75.0 | 78.0 | 60.0 | 91.0 | 63.0 | | |
| PA | 82.0 | 42.0 | 47.0 | 83.0 | 87.0 | 0.65 | 80.48 |
| F1-score | 78.0 | 54.0 | 52.0 | 87.0 | 73.0 | | |
| | | | 2016 | | | | |
| UA | 85.0 | 84.0 | 38.0 | 90.0 | 80.0 | | |
| PA | 86.0 | 54.0 | 37.0 | 89.0 | 85.0 | 0.75 | 85.43 |
| F1-score | 85.0 | 66.0 | 38.0 | 89.0 | 82.0 | | |
| | | | 2018 | | | | |
| UA | 96.0 | 75.0 | 61.0 | 95.0 | 79.0 | | |
| PA | 84.0 | 67.0 | 40.0 | 89.0 | 96.0 | 0.80 | 88.71 |
| F1-score | 90.0 | 70.0 | 48.0 | 92.0 | 87.0 | | |
| | | | 2020 | | | | |
| UA | 80.0 | 86.0 | 50.0 | 89.0 | 89.0 | | |
| PA | 89.0 | 32.0 | 33.0 | 94.0 | 87.0 | 0.76 | 86.86 |
| F1-score | 84.0 | 47.0 | 40.0 | 92.0 | 88.0 | | |

NVA = non-vegetated areas; BA = built-up areas; CL = croplands; OEDF = open evergreen and deciduous forest; DV = dense vegetation; UA = user accuracy; PA = producer accuracy; and OA = overall accuracy.

The accuracy of the classification for croplands and built-up areas was relatively low when compared with the overall accuracy observed in several mappings of this region carried out with satellite data of medium resolution and in different scales due to the peculiar characteristics of these classes. The overall accuracy calculated over these five years was slightly above the mean accuracies that were assessed when using Landsat data [34].

The lowest F1-scores were related to the cropland areas: 38.0 and 40.0 for 2016 and 2020, respectively, and the built-up areas. These low accuracies were, to some extent, not surprising, given their distribution patterns on the ground (Figure 6) and the imbalance of training samples between classes [76].

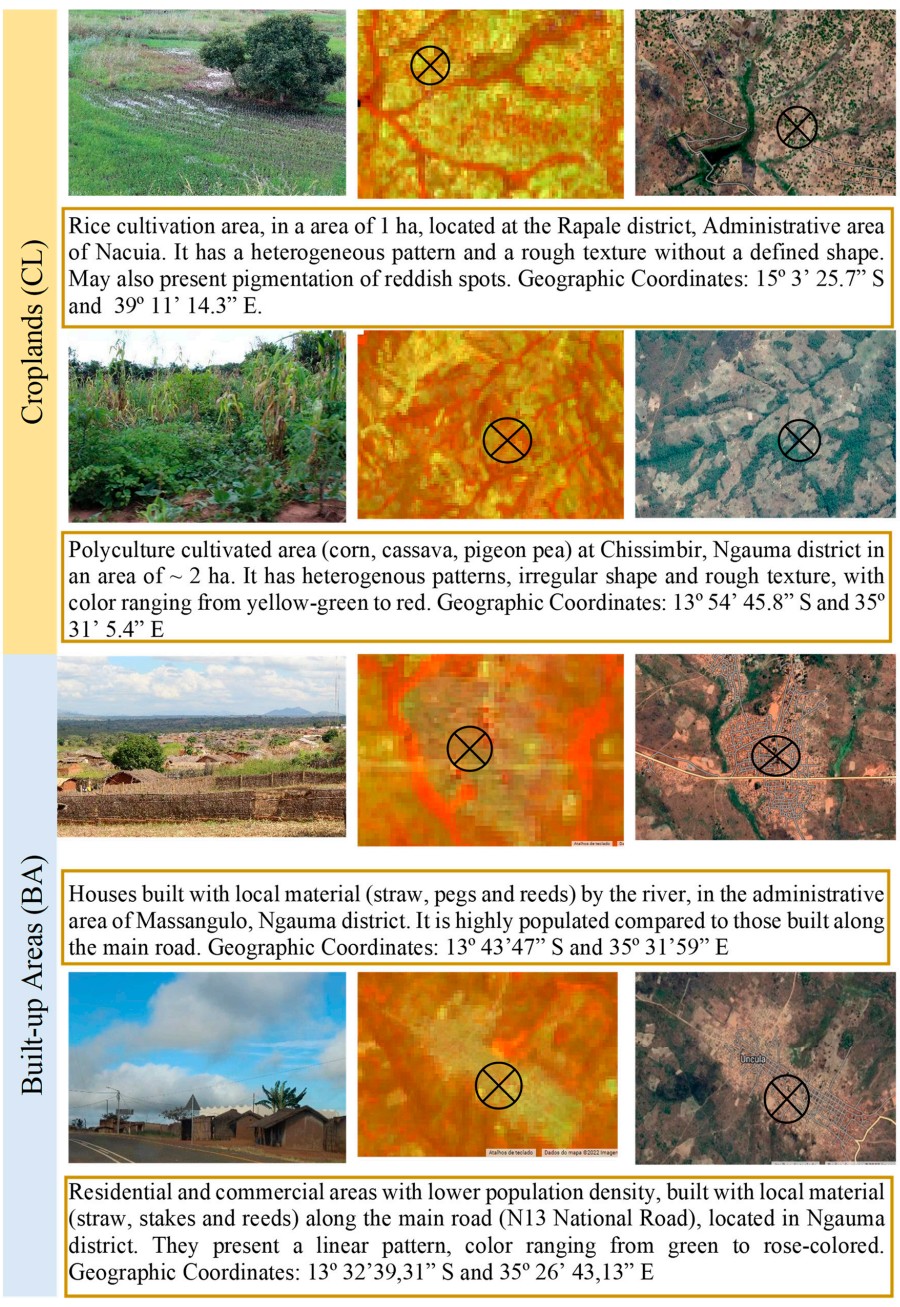

**Figure 6.** Main characteristics of the agricultural areas and built-up/urban areas in the northern region of Mozambique observed in Landsat and high-resolution images. The symbol present in the images in the middle column and on the right indicate the exact location of the area shown in the photo on the left.

### 3.4. Land Use and Land Cover in the Northern Region of Mozambique

As reported before, the following LULC classes were mapped in this study: non-vegetated areas, built-up areas, croplands, open vegetation (savanna), and dense vegetation (forest) (Figure 7). According to these maps, a large extension of non-vegetated areas was confused with built-up area, mainly in the Niassa province, which is the province with the lowest population density in the region and has a limited urban structure. Phan et al. [47] stated that this low accuracy may be associated with the dataset selection performed in this study, which is a crucial step in the classification using GEE in areas with similar dynamics.

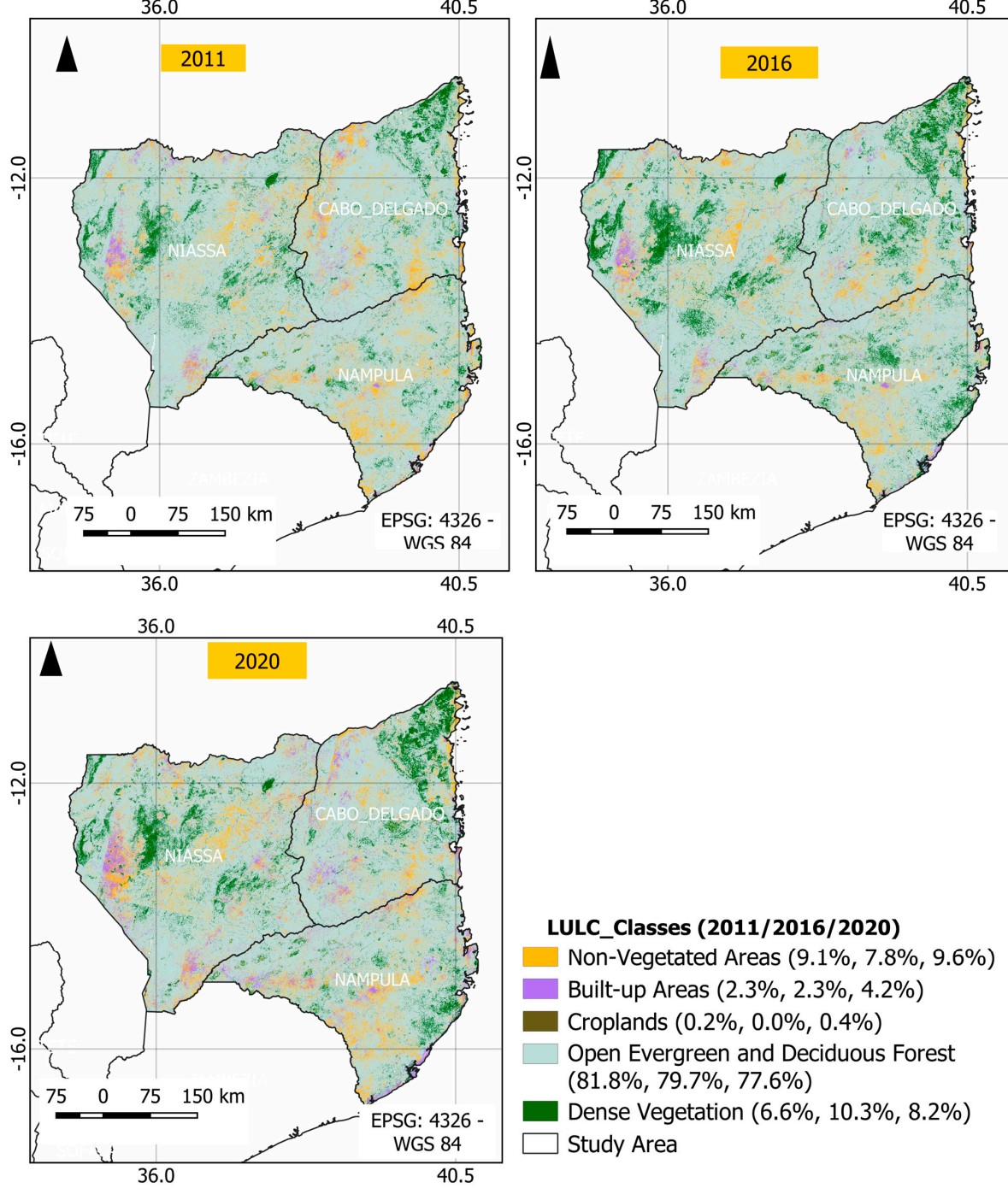

**Figure 7.** Land-use and land-cover maps of the northern region of Mozambique, years 2011, 2016 and 2020. Proportion of each class for each year is shown in the legend.

Although this region is among the most favorable in agricultural terms, this activity is not so expressive. Only small fragments were observed in the provinces of Nampula and Niassa (Figure 7). We can note a predominance of open evergreen and deciduous forest in its different substrates (tree, shrub, and grass), followed by non-vegetated areas, dense vegetation, built-up areas, and croplands, despite some alternations observed in certain years. This result is consistent with mappings performed by Oliveira et al. [77] and ESA (https://2016africalandcover20m.esrin.esa.int/download.php, accessed on 5 July 2022) which indicated an area of native vegetation surpassing 60% in this region.

The Niassa province showed higher extension (47%) than that of Cabo-Delgado and Nampula, with (30%) and (22.9%), respectively. Considerable transitions were verified between dense and open evergreen and deciduous forests. The remaining classes did not present major impacts.

The graphics in Figure 8 show the proportions of surface area of transitions between classes for years 2011 to 2016, 2016 to 2020, and 2011 to 2020. In general, LULC change detection between 2011 and 2020 revealed that non-vegetated areas had increased by 0.7%, built-up by 2.0%, and dense vegetation by 1.3%. On the other hand, open evergreen and deciduous forests had decreased by 4.1% and croplands by 0.01%. The dynamics between 2011, 2016, and 2020 highlight a 3.3% higher gain than loss between 2011 and 2016—a scenario that was opposite to that between 2016 and 2020, in which the loss was approximately 2.4% higher than the gain. Of the total 50,410 km$^2$ of dense vegetation in 2011, approximately 8,261 km$^2$ became open vegetation in 2016. In 2016, approximately 14,580 km$^2$ of dense vegetation became open vegetation, in contrast to an open vegetation gain of 8,945 km$^2$. However, the analysis of the dynamics between 2011 and 2020 showed a higher gain than loss (4.7% versus 3.7%), which revealed a vegetation stability scenario, as shown in Figure 9.

Between 2011 and 2020, 9285 km$^2$ (3.2%) of forest was converted into savanna, most likely because of shifting of cultivation and fuelwood extraction—an activity on which a large part of the population of this region has been dependent for a long time [13,78]. According to CIFOR [79] and Magalhães [80], the Miombo forest is the main source of livelihood for the rural population.

The province that recorded the highest forest loss between 2011 and 2020 was Niassa (4.3%), accounting for 45.9% of the total forest loss, followed by Cabo Delgado (4.1%) and Nampula (3.8%). Conversely, the province with the largest forest gain was Cabo Delgado (6.2%), accounting for 31.1% of the total forest gain, followed by Nampula (5.6%) and Niassa (4.9%). In turn, a proportional forest restoration area of approximately 13,919 km$^2$ (4.7%) mainly consisting of savanna, accounted for approximately 4.4%.

The forest and savanna areas were cyclically transformed in an alternating sequence, that is, on the one hand, the forest became savanna and, on the other hand, the savanna became forest. Figure 10 shows the temporal and spatial LULC dynamics of the period 2011–2020 in the highly forested provinces of Niassa and Cabo Delgado, respectively, and the spatial transitions of the forest in these provinces.

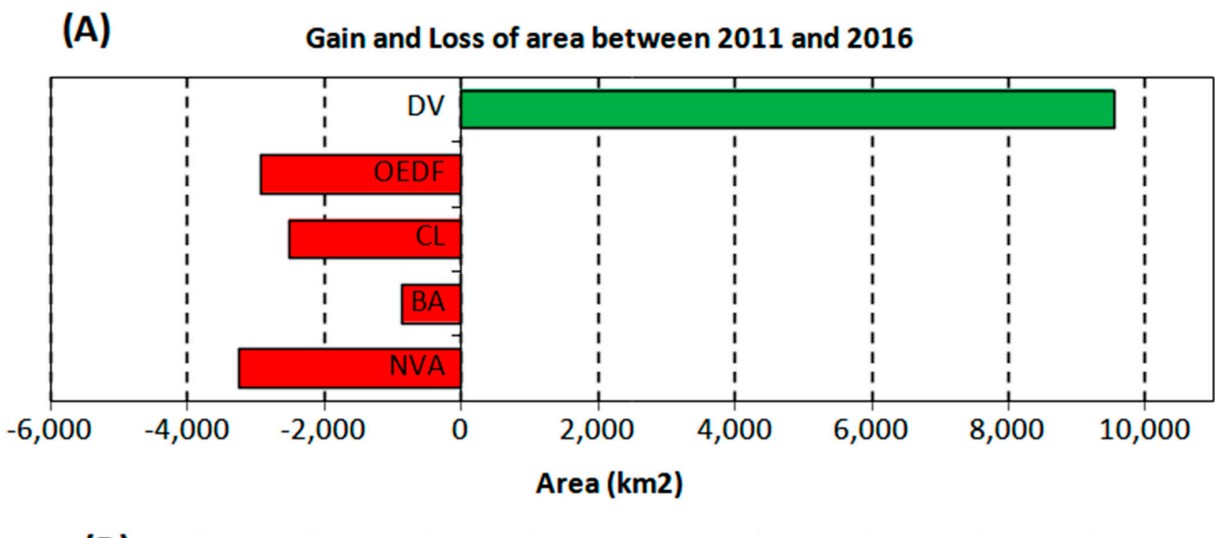

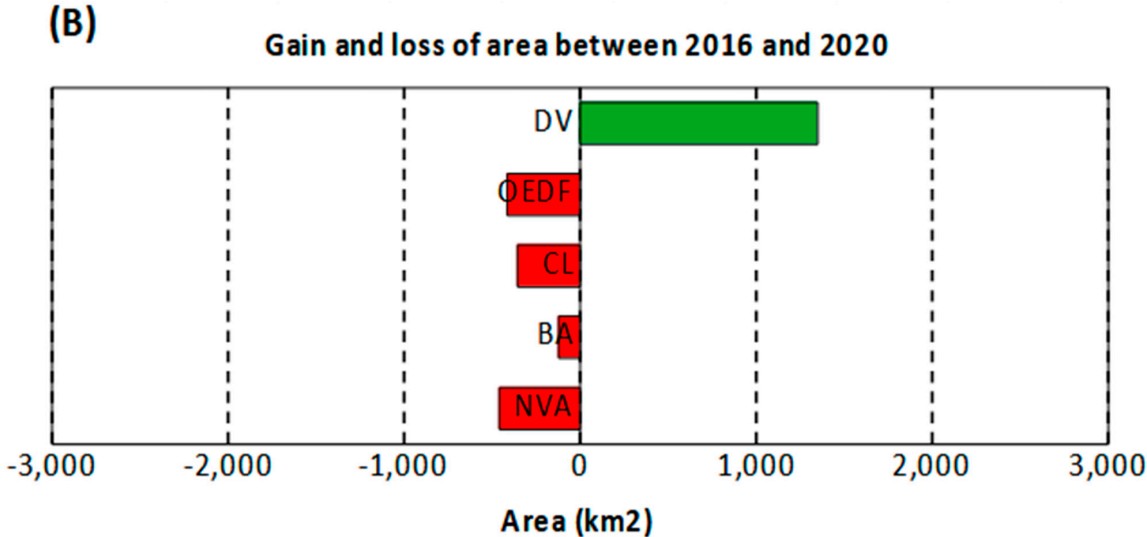

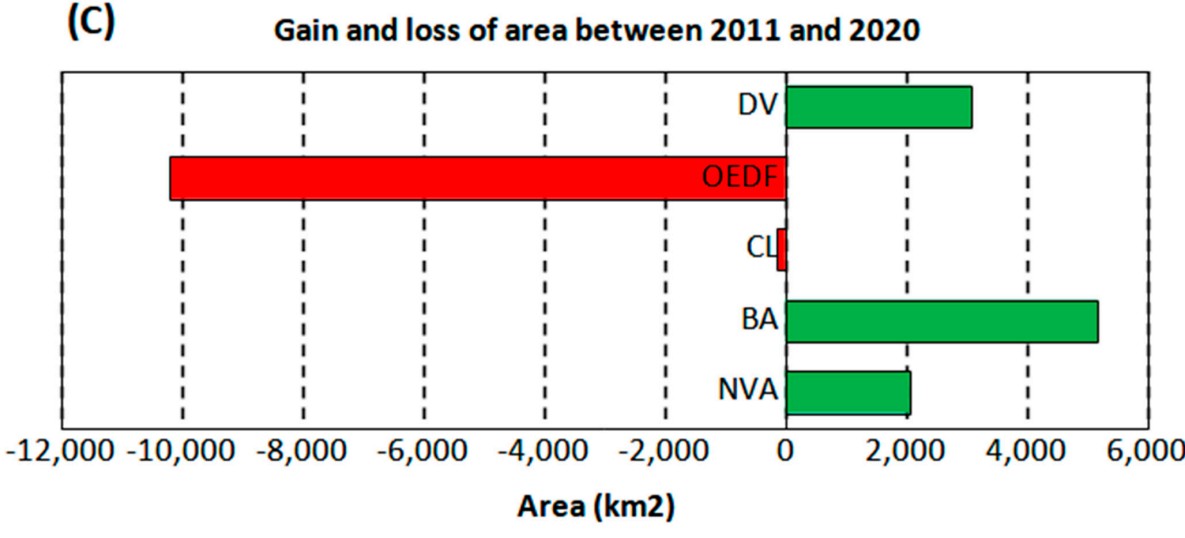

**Figure 8.** Graphics showing land-use and land-cover transitions between 2011 and 2016 (**A**), 2016 and 2020 (**B**), and 2011 to 2020 (**C**). DV = dense vegetation; OEDF = open evergreen and deciduous forest; CL = croplands; BA = built-up areas; and NVA = non-vegetated areas.

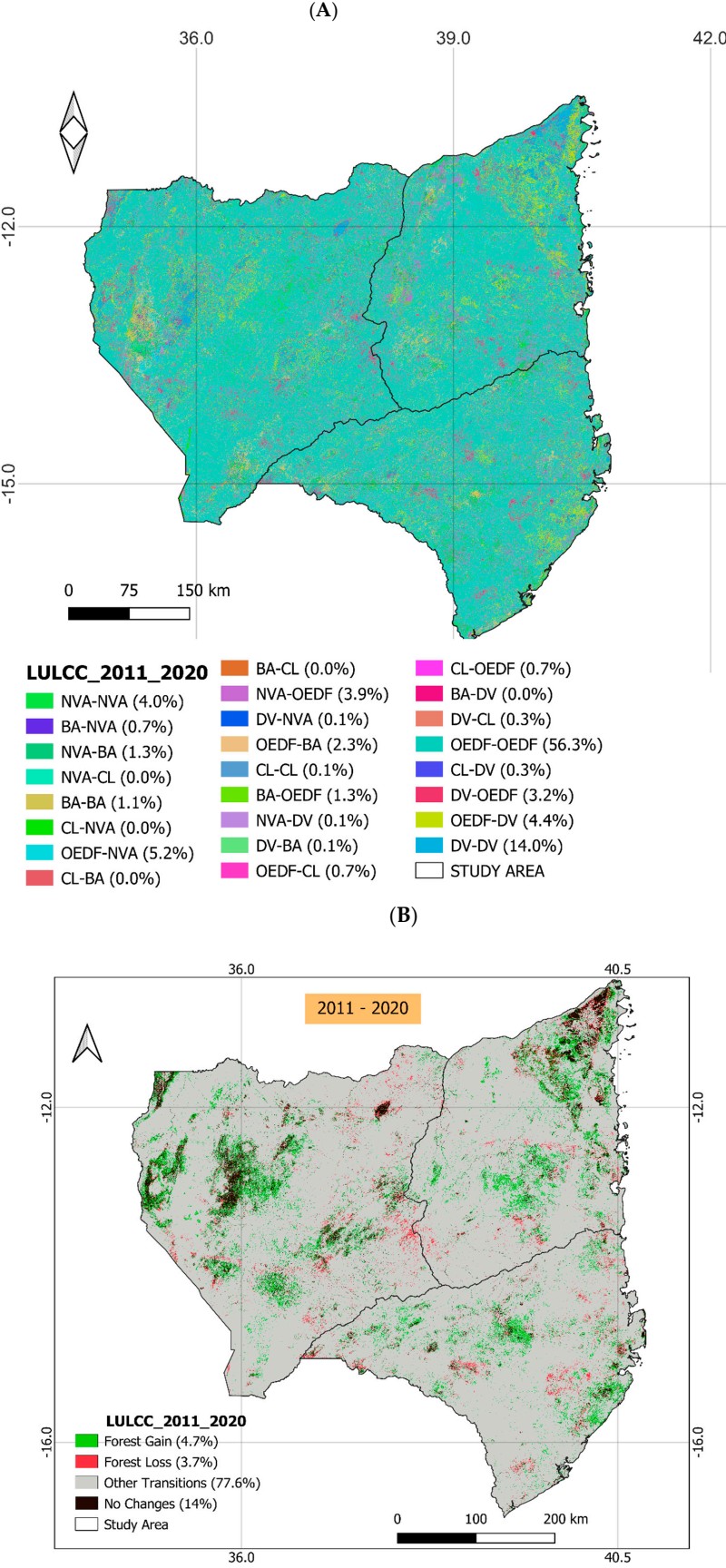

**Figure 9.** Changes in land use and land cover between 2011 and 2020 showing all transitions (**A**) and highlighting the forest (dense vegetation) transitions (**B**).

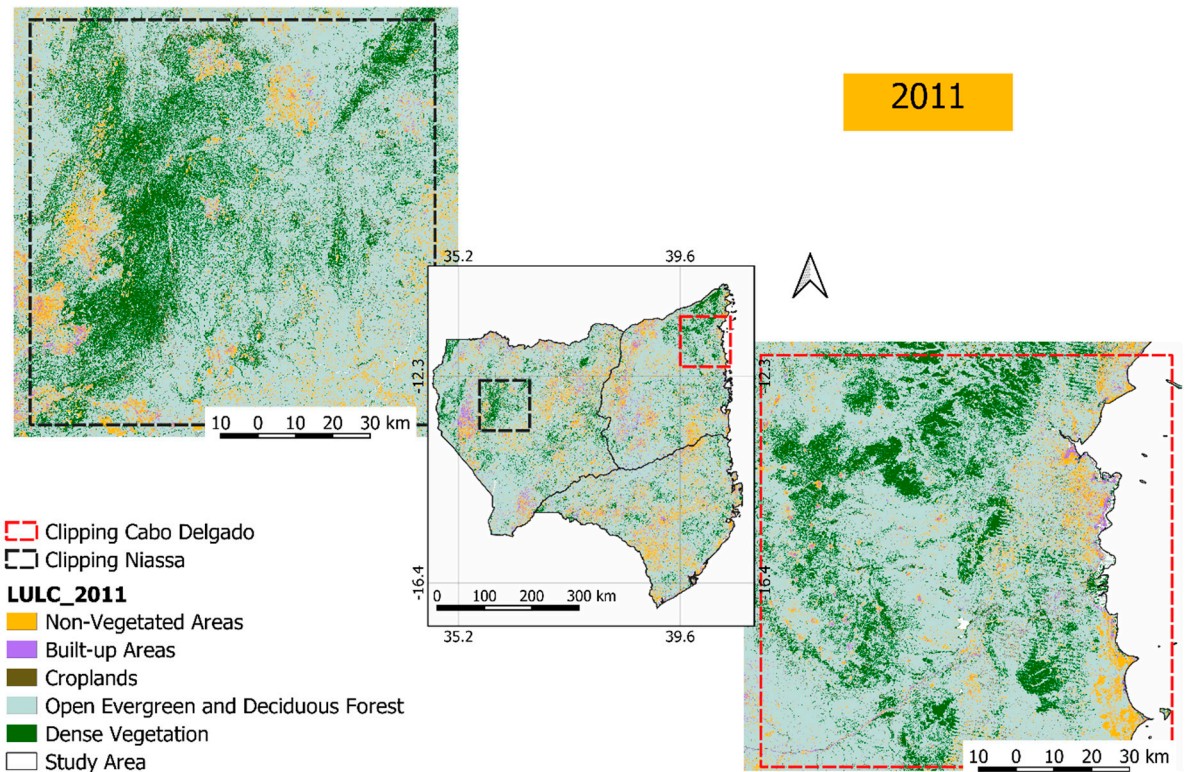

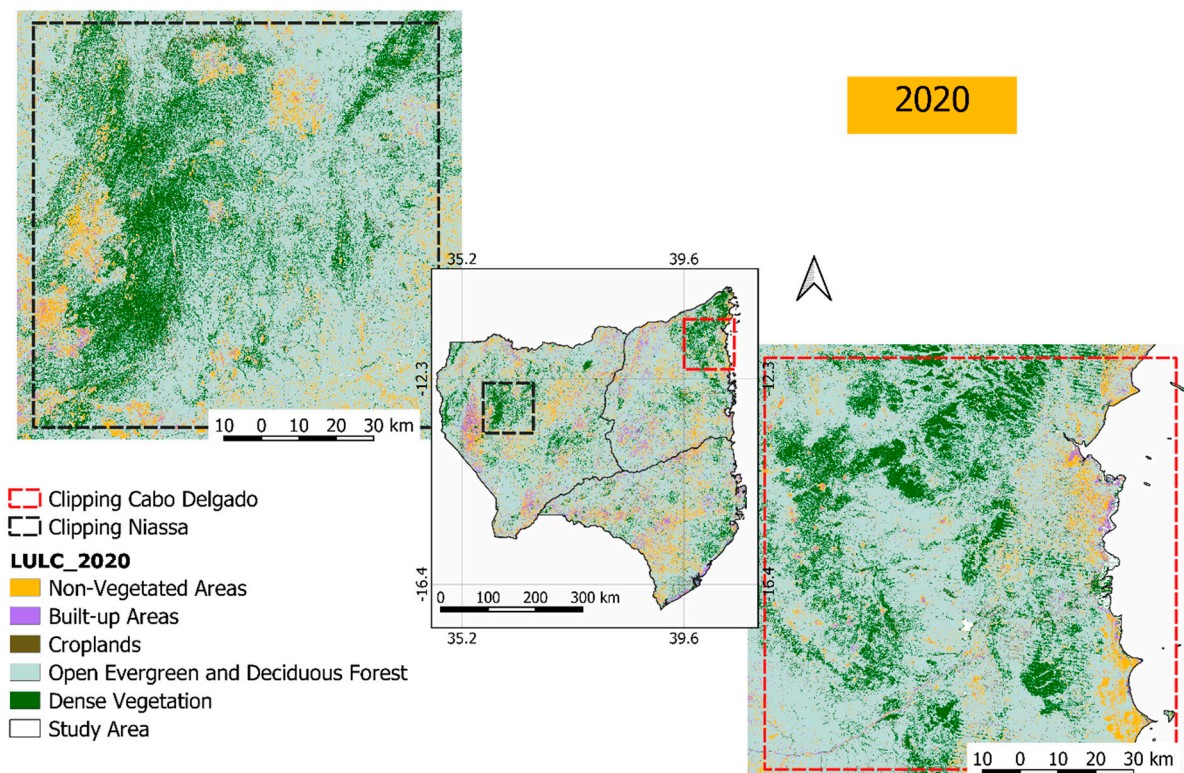

**Figure 10.** *Cont.*

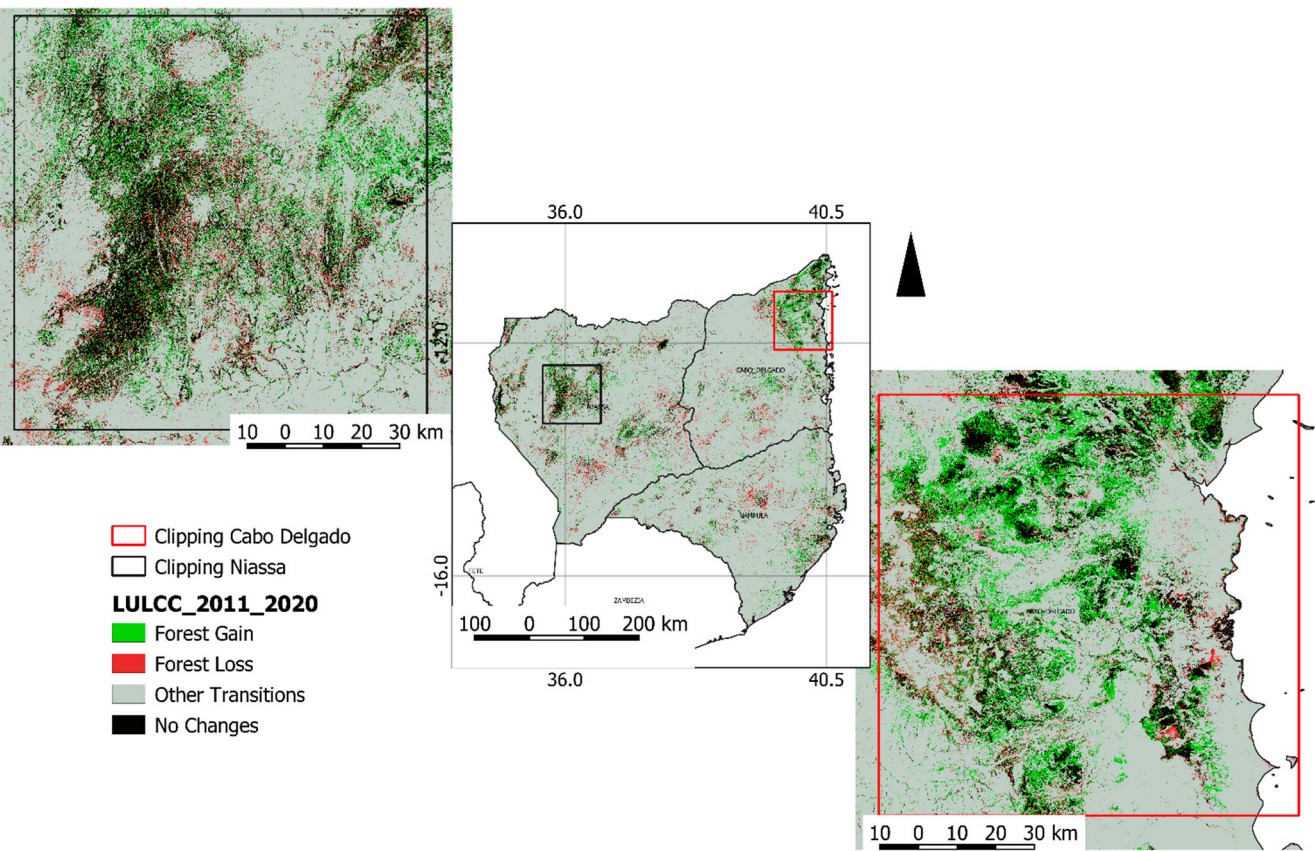

**Figure 10.** Spatiotemporal classification of land use and land cover in the northern region of Mozambique and changes in forest cover between 2011 and 2020, highlighting areas with increased forest expression and forest-related transitions.

## 4. Discussion

The combination of sample-collection techniques and the proportion of training and validation samples [29,35], temporal aggregation using the median, and categorization of the variables by order of importance [35,47] provided high classification accuracy, mainly for primary and secondary vegetation. The RF algorithm proved adequate in terms of data redundancy and classification processes, reducing the processing time significantly. RF in FS techniques allowed assessment of the importance of spectral indices and training-sample size and the quality of the classification, as highlighted by Shetty [33] and Lei et al. [35].

Although the literature indicates that commonly used indices such as the NDVI, EVI, and SAVI, usually improve the accuracy of LULC classification based on satellite images [45,47], in this study, except for NDVI, these indices had little relevance—some of them were even excluded from the classification because of their lack of relevance, which was the case for EVI. The mNDWI also proved to be of little relevance, possibly because of its limitation in separating water from shaded surfaces, typical of the regions with "inselbergs" distributed throughout the region.

In this study, different from what was observed in Duro et al. [40] in which the features showed little difference in terms of importance, we found significant differences between their scores in terms of importance, demonstrating a significant correlation reduction between them. Bessinger et al. [45] suggested that the number of variables used as input has a greater influence on model accuracy than the number of trees used or the number of variables used per split. This scenario was consistent with our results in which a higher number of trees, on the contrary, required more processing time.

Zhang and Yang [81] compared thematic accuracies of LULC maps generated by different models (bands only; all features; and the best-fit). They observed better performance

with the model that used only the best-fit features. Our best-fit model reached 90% to 96% of accuracy for the entire time series.

Phan et al. [47], analyzing the effect of different composition methods and different input images on the classification results, found that if only spectral feature bands from L8 are used, a moderate to high agreement with reference data could be achieved (from 77.6% to 85.27%). However, when additional variables were included in the model, the overall accuracies increased by approximately 4.1% to 7.7%. On the other hand, tests carried out in different seasons showed significant differences in overall accuracy.

Hu and Hu [75] pointed out that poor balancing of the samples can cause low accuracy. Naboureh et al. [76] proposed integrating the subsamples of most classes into a set of vector machines named random under-sampling ensemble of support vector machines (RUESVMs).

The thematic precision of the mapped LULC classes, measured from the user's perspective, revealed that the agricultural areas generated considerable confusion between the open and dense vegetation classes, as well as between urban areas and non-vegetated areas and open vegetation classes. This result was also reported by Oliveira et al. [77] who mapped the same region, and by Phan et al. [47]. These confusions are particularly common in this region, given its highly fragmented agricultural areas with mixed crops on plots with average surface area of 2 ha [31,70,77].

Pullanikkatil et al. [74] stated that, during the dry periods when there is little photosynthetic activity, grazing causes bare-soil exposition in remaining vegetation, resulting in similar spectral values, making it difficult to distinguish between cultivated areas, woodlands, and shrubs.

Differences in accuracies generated in this research can be associated with the calibration of ETM+ and OLI sensors at the level of TOA. Roy et al. [82] found that, on average, the TOA reflectances from OLI were higher than those from ETM+ in all bands, with the largest differences in the NIR and the SWIR bands due to the differences in the spectral response functions between the sensors.

Ronquim et al. [30] emphasized that excessive agricultural land occupation with basic food crops (corn and cassava) in small areas (2 ha, on average) leads to the formation of extensive indistinct agricultural landscapes, which are often interspersed with native savanna vegetation. The same indistinct landscapes occur in built-up or urban areas, most of which are characterized by unpaved roads and houses covered with straw or reeds and zinc sheets, which are confused with non-vegetated areas (Table 5).

The largest agricultural area visited during the fieldwork had an extension of approximately 18 ha. The small-scale farming in the region is associated with institutional constraints such as the lack of sustainable agrarian policies, lack of financing for smallhold farmers, inequalities in accessing agricultural credits, reduced availability of rural workers, and failure to comply with land use, among others, limiting the development of large-scale agriculture [83]. Regarding non-compliance with land-use plans, a study conducted by Bey and Meyfroidt [83] indicated that, from 2001 to 2017, approximately 70% of the expansion of large-scale tree plantations occurred on agricultural lands. Over 40% of the expansion of plantations occurred on land illegally designated for that use. Other limitations were related to the spatial resolution of the Landsat ETM+ and OLI sensors in detecting smaller areas.

The LULC map from 2020 showed that the entire study area presented about 117,000 hectares of croplands, that is, 0.4% of the total area of interest. In terms of forestlands, we found 25 million hectares, that is, 86% of the study area. These results are quite different from those published in the Atlas of Mozambique Forest Resource Reference Map of 2013 [84] 458,000 hectares of croplands (1.6%) and 17 million hectares of forestlands (60%). Several factors can explain these differences in area estimation, especially the differences in methodological approaches and legend as well as the time lag of seven years between two mappings.

A complementary study should be performed with an improved spatial resolution to better separate the classes that were grouped in this research, for example, mangroves and primary vegetation, to provide more detail about the transitions that occurred in the region. Jew et al. [85] and Fleischman et al. [86] argued that the woody and herbaceous vegetation of the Miombo has a high capacity for natural recovery after disturbances due to agriculture, charcoal production, logging, and fires.

In the analysis of transitions, the agents are usually difficult to identify in Mozambique. According to Sitoe et al. [13], these difficulties result from the complexity of the interactions between agents. These authors identified the following multiple systems of agents: commercial and shifting agriculture, firewood, charcoal, urbanization, mining, logging, and livestock. Many of the direct causes act simultaneously and are difficult to separate. Moreover, they can act in the same region in combination, either in the same period or consecutively.

The complexity in identifying drivers of LULC change was also found in other studies. Pullanikkatil et al. [74] identified the demand for agricultural land as a result of population growth as the main driver of LULC change. Hu and Hu [75] reported the effects of increasing urban populations and expansion of industries as other drivers.

Although it was not the objective of this study to identify drivers of land-cover change, over the period defined in this research (2011–2016 and 2016–2020), previous studies have indicated that these drivers were related to firewood and charcoal extraction, logging, and cultivation shifting [13,87]. Guedes et al. [87] stated that the income of rural families derives from the sale of firewood and charcoal extracted from adjacent forests. Additionally, these resources are used for food, construction, medicine, and work tools. CIFOR [79] and Magalhães [80] believed that the high dependence of rural populations on forest resources is the major factor in high deforestation and forest degradation rates, which are exacerbated by their fragility.

In their study on mapping of smallholder and large-scale agricultural-land dynamics with a flexible and composite pixel-based classification system in an emerging frontier of Mozambique, Bey et al. [12] identified considerable areas of other classes converted into small-scale agriculture, supporting the assertion that this activity contributes to deforestation. Sitoe et al. [13] revealed that shifting cultivation plays a key role in the deforestation process. Guedes et al. [87] stated that when an area experiences a land-cover change as a survival measure, rural populations migrate to areas that are still productive. Because their livelihood depends on sedentary agricultural practices, they change the LULC.

Notwithstanding the relative forest-cover stability observed during this period, the type of forest that is being suppressed must be closely studied because forests are crucial for biodiversity conservation and ecosystem services [87]. In the 10-year analysis performed in this paper, no alarming deforestation scenario was detected. Therefore, the results do not corroborate the findings that have been reported in media and in the National Forestry Inventory of 2018 [3]. This discrepancy does not imply that those findings should be questioned, but rather that such information reported in media be based on spatial data and that those institutions responsible for the National Forestry Inventory develop uniform mapping methodologies for spatiotemporal analyses.

The method proposed in this research has the advantage of being able to process large volumes of data. The method also allows reduction of data dimensionality by excluding highly correlated data sets, consequently reducing processing time. Another advantage is the possibility of handling several types of variables with different units of measurement, for example, spectral bands, spectral indices, textural attributes, and digital elevation models. The main limitations of the proposed method are two-fold: (i) its success is strongly dependent of the quality of ground truths and (ii) there is a relatively strong demand for computational capability to run ML algorithms whenever the volume of data is big. However, this limitation can be bypassed if the analyst has access to a cloud-processing facility and has knowledge of language programming.

## 5. Conclusions

This research involved dense data derived from Landsat 7 ETM+ and Landsat 8 OLI satellites on an area covering 24 images. The RF algorithm was shown to be suitable for analyzing and testing the model for the selection of variables. The use of spectral indices and their ranking by importance, the data reduction using the median, and the determination of the number of classes for classification proved relevant in this study by providing good accuracy considering the characteristics of the area and the volume of processed data. These procedures significantly reduced the volume of data, facilitating and accelerating the analysis.

The forest cover in the study area was relatively degraded and stable. Minor changes were related to conversion to savanna and vice-versa. This needs to be monitored on a regional or national scale to avoid the advance of this scenario. Therefore, the development of more studies is strongly encouraged.

The workflow presented in this study revealed the relevance of the data dimensionality reduction step in LULC classification, opening possibilities for exploring other methods based on both FS and feature extraction. Other key components that required a detailed analysis were the definition of the number of classes and the size of the samples. Similar studies conducted in this region should define the classes and sample sizes carefully.

The RF algorithm and the GEE platform were suitable for FS and classification processes, reducing the processing time considerably. A few processing trials of the same data in other geographic information systems and digital image-processing software proved inappropriate because they were not completed within the required time.

Complementary studies with additional data and techniques, such as textural attributes extracted from the gray-level co-occurrence matrix (GLCM), as well as other spectral, spatial, and temporal indices and other feature selection methods, should be tested and combined with time-series analysis to improve classification accuracy.

**Author Contributions:** Conceptualization for the overall paper and writing the Abstract, Introduction, development, and Conclusions, Lucrêncio Silvestre Macarringue; review, data analysis, and editing, Édson Luis Bolfe; script design and data analysis, Soltan Galano Duverger; review and editing, Edson Eyji Sano, Marcellus Marques Caldas, Marcos César Ferreira, Jurandir Zullo Junior and Lindon Fonseca Matias. All authors have read and agreed to the published version of the manuscript.

**Funding:** This research was funded by National Council for Scientific and Technological Development (CNPq), grant number 190158/2017-4, and by Fundo Nacional de Investigação of Mozambique, grant number unavailable.

**Institutional Review Board Statement:** Not applicable.

**Informed Consent Statement:** Not applicable.

**Data Availability Statement:** Not applicable.

**Acknowledgments:** The first author thanks CNPq for the doctoral scholarship as part of the project of monitoring land use and land cover in northern Mozambique. He also thanks to the Fundo Nacional de Investigação of Mozambique for funding the field work in 2020. We are thankful for the comments from three anonymous reviewers who helped improve the quality of this paper significantly.

**Conflicts of Interest:** The authors declare no conflict of interests.

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
