# Peer review of "Land Use and Land Cover Classification in the Northern Region of Mozambique Based on Landsat Time Series and Machine Learning"

_ijgi, doi:10.3390/ijgi12080342_

Round 1
Reviewer 1 Report (Previous Reviewer 1)
Thank you for the invitation to review the resubmitted manuscript. I have thoroughly examined the revisions made by the authors in response to my previous concerns and suggestions. After careful consideration, I am pleased to note that the authors have made a commendable effort in addressing these issues. Consequently, I have no further comments to add at this stage. The manuscript may now proceed to the next steps of the review process. I do not require another revised version. Best regards.
Author Response
Dear Reviewer,
Thank you for reviewing our paper.
With best regards.
Reviewer 2 Report (Previous Reviewer 2)
Dear authors,
Manuscript is prepared good, but still need some improvements.
Section 2.3.1 – Provide samples distribution between classes. In case of using machine learning models, it is crucial.
Coloring on Figures 7 and 10 must be changed to another colors. It is hard to distinguish difference between classes.
I think it will be a good idea to compare achieved results with other LULC sources, because they’re representing different results, especially in classification agricultural areas (ATLAS OF MOZAMBIQUE FOREST RESOURCE REFERENCE MAP).
Generally, the main problem of this research, in my opinion, in class selection. From Figure 5 there’s no significance difference between classes can be seen. Authors should think about classes not from land use point of view, but from the reflectance of the materials these classes consist of. Thus, classes are hardly different one from another (Table 5), which lead to results from page 18.
“Despite these difficulties identifying drivers of change, for the period defined in this research (2011–2016 and 2016–2020), this result suggests that the change is caused by firewood and charcoal extraction by local communities, logging, and shifting cultivation.” – in my opinion this sentence should be removed in present form or modified. Defining the drivers of the LULCC is a completely different task and research. In present form it is just an ungrounded authors suggestion.
I wish that my comment would be helpful in improving the quality of this research.
Thank you.
Author Response
Dear Reviewer,
Thank you for your valuable comments. Find our respondes in the attached document.
With best regards.

Reviewer 3 Report (New Reviewer)
Please see the attachment.
The manuscript entitled "Land Use and Land Cover Classification in the Northern Region of Mozambique based on Landsat Time Series and Machine Learning" In this manuscript time-series Satellite datasets are used to map land surface features using machine learning classification (i.e., Random Forest Classifier) method. This manuscript is written well and the drawn conclusions are coherent with the obtained results. Although similar methodologies are common, the provided information could be useful for improving land management actions.
Comments and suggestions:
1. Introduction:
- " Mozambique has enormous natural resources (forest) " Please revisit this expression; I suggest focusing on forest or land surface feature not natural resources.
- Please outline the objectives of the study at the end of the introduction.
2. Materials and Methods
- " It was necessary 24 Landsat scenes per year to cover the entire study area." Please refine this sentence.
- " The hyper parameters used in the RF were the numberOfTrees and the" please refine this expression.
4. Discussion
- A small section highlighting the benefit and limitations of the applied classification technique and its transferability is recommended.

Author Response
Dear Reviewer,
Thank you for your valuable comments. Find our respondes in the attached document.
With best regards.

Round 2
Reviewer 2 Report (Previous Reviewer 2)
From the previous version of the manuscript authors significantly improved it. Additional thanks for good color scheme for figures. Currently all classes can be differ.
In my opinion, manuscript can be published in present form
This manuscript is a resubmission of an earlier submission. The following is a list of the peer review reports and author responses from that submission.
Round 1
Reviewer 1 Report
Thank you for the invitation to review this manuscript on investigating LULC changes in Northern Mozambique using Landsat time series and random forest algorithm. The topic is intriguing, and the paper is well-organized. However, I believe there is room for improvement, and I recommend the following major revisions:
- Clarify why you chose to use an external coding for the random forest classifier instead of GEE's built-in RF classifier. Additionally, please indicate which Python packages you used for training the RF classifier since Google Colab is a coding environment, not a package.
- Explain how you optimized the hyperparameters of the RF classifier and define the technique used for optimization.
- Provide more justification and explanation for the computation of feature importance. Please explain why you needed to compute the contribution of each index to the learning phase and provide a clear explanation of your base.
- Simplify the Sankey diagram to improve its readability. Alternatively, consider using bar plots.
- Justify why you chose to show only forest gain and loss in your transition maps instead of displaying all transition types.
- Provide more details on the GEE and Landsat data sets in the methodology section.
- Justify why you chose to use the random forest classifier and explain its advantages compared to other classifiers.
- Compare your performance with similar and novel studies using GEE, Landsat, and RF. Please do not compare your findings with studies using Sentinel-2 and MODIS. You can refer to recommended readings for guidance.
- Re-organize your conclusions without bullet points and numbering.
Overall, the findings are sound, but the paper requires major revisions to enhance its clarity, rigor, and impact.
Recommended readings to refer:
https://doi.org/10.1007/s10661-022-10437-6
https://doi.org/10.1080/09640568.2021.2001317
https://doi.org/10.1016/j.sajb.2022.08.014
https://doi.org/10.3390/rs14133191
https://doi.org/10.1007/s12517-022-10158-7
Author Response
Dear reviewer 1.
Thank you for your valuable comments to our article. They were useful for the improvement of it.
We agreed with many of you comments and we made changes in the paprer.
Best regards.

Reviewer 2 Report
The effort LULC classification using GEE and machine learning in region is quite interesting and will bring a significant contribution in this field.
Besides this, the manuscript needs some improvements.
In my opinion the goal of the research should be transformed into something more important. The goal “mapping LULC classes between 2011 and 2020” looks too simple.
Subsection 2.2. It is a good idea to specify dataset used in this study, from GEE catalog (LANDSAT/LE07/C02/T1_TOA for example). Also, it is good to specify bands, used in this research.
In my opinion, the main problem of this manuscript is class selection. From the provided results, seems it is necessary to exclude OEDF, due to high similarity with other classes based on Figure 4, Figure 6. According to the Figure 8, this class occupies ~ 80% of the research area, but based on its similarity with other classes, final results can significantly differ from reality. Also, from the table 1 (field photos) seems there is no difference between OEDF, grassland and croplands from the RS point of view. All of this classes have similar type of vegetation with same amount of the chlorophyll, thus it will be hard to separate one class from another.
I wish that my comment would be helpful in improving the quality of this research.
Thank you.
Author Response
Dear reviewer 2.
Thank you for your valuable comments to our article. They were useful for the improvement of it.
We made changes in the paprer.
Best regards.

Reviewer 3 Report
The manuscript by Macarringue et.al investigates the Land cover classification and change in northern part of Mozambique through applying Random Forest models and some spectral-related variables. Although the manuscript is well structured, to some extent, there are still several issues that avoid me to suggest it for publication. Therefore, I suggest major revision.
1- what you mean by “Dense vegetation gain exceeded loss by 1.5%, while 3.4% of the study area, classified as dense vegetation, did not change during this period. The remaining 88.7% are related to other transitions not considered in this study.” The whole sentence is not clear, so revise it completely and explain what you mean as well.
2- Please clearly explain the importance of your study for other regions than Mozambique. What are the novelties. This can be explained in abstract and also last paragraph in introduction.
3- Please revise your keywords: I do not think cloud computing is related to this manuscript.
4- In introduction try to expand your literature review specially by mention the papers published in IJGI. Also to emphasis more on the importance of LULC for sustainable development and importance of machine learning techniques in image processing cite the following useful references:
10.1109/ACCESS.2022.3163535
https://doi.org/10.1016/j.chemosphere.2022.135835
10.3390/app7060566
5- Please enhance the resolution in Fig.1. If possible, use a better base map.
6- I can see in Fig.1 some areas are filed only by test samples while some which training, would you please explain how to divided your samples to test and train. By this distribution, maybe your result is not trustable.
7-Provide the total number of test and training samples in Fig.1
8-Section 2.2. is written too briefly. Please provide a comprehensive explanation about the data set, including the spatial resolution, temporal resolution, which bands were used, etc.
9-Some of the steps in Fig. 2 were not explained in the text, such as post-classification?
10-What is smile. RF?
11-You mentioned that you used RF for feature selection. But the RF importance scores are only based on the role of different variables in splitting nodes. Therefore, it cannot give us the rule of thumbs that only these variables that you selected are the most important one. Maybe if we add other variables to current dataset, we could get different results. So how you could explain your variable selection stage just based on importance score.?
12-In Table 2. Provide a threshold for each of the indices. How you use these indices to extract different class. Need threshold.
13-Modify Eq.2.
14- Please enhance the resolution of all figures in the manuscript (gloomy)
15- How you explain the variations in accuracy results? Why the best is 2020 and the worst is 2013?
16- Please stop using abbreviation in the text, what you chose is confusing, causing readers go up and down to find the full name. revise it.
17- what is your main point in Fig.7?
18- There is a big difference between your result of dense vegetation and National forest result. How you explain this and which one we should refer? Almost 100 km2 difference is very big gap.
19- Replace Fig.9 with a better and more clear Fig. why there is no unit in this figure. Please replace it with more understandable figure.
20 – From Fig.10 what I can see is considerable Forest Loss, but in the text you mentioned it is not that huge. How you explain this. Also please add area in square km instead of only percentage.
21- In conclusion, statements 3 and 4 deny each other.

Author Response
Dear reviewer 3.
Thank you for your valuable comments to our article. They were useful for the improvement of it.
We agreed with many of you comments and we made changes in the paper.
Best regards.

Round 2
Reviewer 1 Report
Thank you for inviting me to review the revised paper. While the paper has undergone revisions (especially the artwork is not better), there are still some issues that need to be addressed to ensure publication in the IJGI. Below are my suggestions:
1) Your response letter states that you computed the feature importance by exporting your train data set. However, feature importance is determined not by how well a feature predicts records that built the model, but rather by shuffling the values of the feature and observing whether the model error increases or remains unchanged. Therefore, I suggest using test/validation samples to compute feature importance.
2) The vegetation indices you implemented might lead to bias in plant status due to seasonality. To mitigate this, it is better to decrease the time-period of image selection, such as by using only summer images.
3) Combining L7 ETM+ and L8 OLI images at the same time might result in different spectral responses, especially in NIR portions. To address this issue, please refer to Roy et al. (2016) for guidance on how to handle this. https://doi.org/10.1016/j.rse.2015.12.024
4) You can mention your hyperparameter optimization technique in the text. Your response in the letter (Randomized Search) is acceptable.
5) Your discussion still lacks content. As I previously mentioned, it would be helpful to compare your study with others that use the same materials. For instance, Ma et al. used UAV imagery, Stroman et al. used Sentinel-1 and 2, and Naboureh et al. addressed the class imbalance problem. It is also essential to compare your classification performance with that of studies using Landsat imagery and the RF algorithm (I already recommended some of the in previous round). Your classification performance is relatively low, and it would be valuable to explore existing literature and provide interpretations. Additionally, you should emphasize the novelty of your feature importance analysis in discussions and explain how it improved our understanding of indice selection (based on the existing body of literature).
Author Response
Dear reviewer.
Thank you for your comments to improve the article. Based on the suggestions given, we made changes on the data analysis and consequently the results had some changes and all of them were incorporated into the text.
Please see the attachment.
Best Regards.

Reviewer 3 Report
The manuscript by Macarringue et.al investigates the Land cover classification and change in northern part of Mozambique through applying Random Forest models and some spectral-related variables. After going through the comments I found that my main concern which is accuracy of the result was not addressed. first the distribution of training and validation samples are not good enough to give a trustable results. Second, there is a huge gap between their results and NFI. Third, some of the answers to comment were not satisfying, eg, comments 11,15 and 18. Therefore, I do not suggest publication of it in IJGI as a high level journal in Geoscience field.
Author Response

(The authors gave the same response as above.)
